# Ammonium-Generating Microbial Consortia in Paddy Soil Revealed by DNA-Stable Isotope Probing and Metatranscriptomics

**DOI:** 10.3390/microorganisms13071448

**Published:** 2025-06-21

**Authors:** Chao-Nan Wang, Yoko Masuda, Keishi Senoo

**Affiliations:** 1Department of Applied Biological Chemistry, Graduate School of Agricultural and Life Sciences, The University of Tokyo, Tokyo 113-8657, Japan; 2Collaborative Research Institute for Innovative Microbiology, The University of Tokyo, Tokyo 113-8657, Japan

**Keywords:** ^15^N stable isotope probing, metatranscriptomics, nitrogen-fixing bacteria, ammonium generation, dissimilatory nitrate reduction to ammonium, denitrification, rice paddy soil

## Abstract

Rice paddy fields are sustainable agricultural systems as soil microorganisms help maintain nitrogen fertility through generating ammonium. In these soils, dissimilatory nitrate reduction to ammonium (DNRA), nitrogen fixation, and denitrification are closely linked. DNRA and denitrification share the same initial steps and nitrogen gas, the end product of denitrification, can serve as a substrate for nitrogen fixation. However, the microorganisms responsible for these three reductive nitrogen transformations, particularly those focused on ammonium generation, have not been comprehensively characterized. In this study, we used stable isotope probing with ^15^NO_3_^−^, ^15^N_2_O, and ^15^N_2_, combined with 16S rRNA high-throughput sequencing and metatranscriptomics, to identify ammonium-generating microbial consortia in paddy soils. Our results revealed that several bacterial families actively contribute to ammonium generation under different nitrogen substrate conditions. Specifically, *Geobacteraceae* (N_2_O and +N_2_), *Bacillaceae* (+NO_3_^−^ and +N_2_), *Rhodocyclaceae* (+N_2_O and +N_2_), *Anaeromyxobacteraceae* (+NO_3_^−^ and +N_2_O), and *Clostridiaceae* (+NO_3_^−^ and +N_2_) were involved. Many of these bacteria participate in key ecological processes typical of paddy environments, including iron or sulfate reduction and rice straw decomposition. This study revealed the ammonium-generating microbial consortia in paddy soil that contain several key bacterial drivers of multiple reductive nitrogen transformations and suggested their diverse functions in paddy soil metabolism.

## 1. Introduction

Rice is the primary food source for more than half of the global population, particularly in Asia [1,2]. Rice cultivation depends on various nutrients, making soil fertility a key determinant of both rice yield and quality. Among these nutrients, nitrogen is essential for plant growth and development, as it enhances crop yield, promotes chlorophyll II formation, and improves photosynthetic efficiency [3,4]. Among the available forms of nitrogen, nitrate (NO_3_^−^) and ammonium (NH_4_^+^) are the primary inorganic sources absorbed and utilized by plants [5]. Rice efficiently absorbs ammonium via root transporters [5,6]. In addition, as a cation, ammonium readily binds to most soil types and is less susceptible to leaching than nitrate, ensuring a more stable and continuous nitrogen supply [7]. Consequently, ammonium-based fertilizers are widely used to improve rice production.

However, nitrogen fertilizer production is resource- and energy-intensive, and excessive application releases harmful gases such as ammonia (NH_3_) and nitrogen oxides (NO_x_), which pose environmental and human health risks [8]. Moreover, only a fraction of the applied nitrogen is effectively absorbed by crops; the remainder leaches into surface water and groundwater, threatening aquatic ecosystems [9,10,11]. These concerns highlight the need to reduce nitrogen fertilizer use and adopt more sustainable agricultural practices.

Paddy soils are frequently waterlogged and anaerobic ammonium is naturally generated through microbial processes, thereby contributing to soil fertility and plant growth [12]. The primary microbial pathways responsible for ammonium generation include nitrogen fixation (N_2_ → NH_3_) and dissimilatory nitrate reduction to ammonium (DNRA; NO_3_^−^ → NO_2_^−^ → NH_4_^+^). Moreover, denitrification (NO_3_^−^ → NO_2_^−^ → NO → N_2_O → N_2_), which shares an initial pathway with DNRA, is also prevalent in paddy soils. The final product, nitrogen gas (N_2_), subsequently serves as a substrate for nitrogen fixation [13,14,15,16,17]. These reductive nitrogen transformation processes—DNRA, nitrogen fixation, and denitrification—interact dynamically to regulate ammonium generation in paddy soils.

Despite extensive research on nitrogen transformation in paddy soils, several limitations remain. Most studies have examined DNRA, denitrification, or nitrogen fixation as independent processes, disregarding their associations within the nitrogen cycle. Additionally, the presence of pseudo-*nifH* genes, which are frequently misannotated as functional *nifH* genes, leads to an overestimation of the nitrogen-fixing microbial abundance [18]. DNRA predominates nitrate removal in paddy soils owing to the anaerobic, carbon-rich conditions that preferentially support DNRA over denitrification; however, its ecological significance has only recently gained recognition [14,19]. Furthermore, many microorganisms involved in these processes are challenging to culture under laboratory conditions because of their specific growth requirements and sensitivity to environmental factors, which limits our understanding of their physiological and metabolic functions [20,21]. Recent advances in molecular biology and bioinformatics, such as high-throughput sequencing, isotope labeling techniques, metagenomics, and metatranscriptomics, have significantly improved our ability to study microbial communities [22,23,24]. However, no single technique can completely capture the complexity of microbial community structure and function, underscoring the need for an integrated approach to comprehensively elucidate reductive nitrogen transformation processes and the microorganisms involved.

This study aimed to identify ammonium-generating microbial consortia involved in reductive nitrogen transformation in paddy soils. To achieve this goal, paddy soil microcosms were established individually using stable isotope labeled nitrate, nitrous oxide, and nitrogen. The production and consumption of nitrous oxide and nitrogen, which indicate the dynamics of the added nitrogen compounds, were monitored via gas chromatography-mass spectrometry (GC-MS). DNA-stable isotope probing (DNA-SIP), combined with 16S rRNA high-throughput sequencing, has been used to identify the microorganisms responsible for assimilatory reactions, such as ammonium or nitrate incorporation and biological nitrogen fixation. Additionally, a metatranscriptomic analysis was performed along with SIP to identify the microorganisms involved in both assimilatory and dissimilatory nitrogen transformations, including denitrification and DNRA.

## 2. Materials and Methods

### 2.1. Soil Sampling and Characterization

Soil samples were collected from an experimental paddy field at the Niigata Agricultural Research Institute, Nagaoka, Niigata Prefecture, Japan (37°44′ N, 138°87′ E), at a depth of 0–20 cm. No nitrogen fertilizer was applied to the field, as previously described [25]. During transport, the soil samples were stored in iceboxes. The samples were sieved through a 2.0 mm mesh to remove impurities and thoroughly mixed in the laboratory. The homogenized soil was then stored at 4 °C for subsequent analyses. The physicochemical properties of the samples are listed in Appendix A.

### 2.2. Nitrate and Nitrous Oxide Concentration Gradient Experiments

The isotopically labeled compounds ^15^N_2_, ^15^N_2_O, and ^15^NO_3_ were used for the SIP analysis. Our previous SIP experiments using ^15^N_2_ established that an optimal ^15^N_2_ concentration of 80% and an incubation period of 72 h are suitable. However, the optimal concentrations for nitrate and nitrous oxide remained undetermined. Therefore, we performed a microcosm experiment using a concentration gradient of nitrate and nitrous oxide. The ambient conditions for SIP analysis were determined by quantifying nitrous oxide and ammonium generation in the nitrate gradient experiment and nitrous oxide consumption in the nitrous oxide gradient experiment. Microcosms were prepared by mixing 5 g of fresh soil with 2.5 mL of distilled water in 20 mL glass serum vials. The vials were sealed with butyl rubber stoppers and aluminum crimps. To deplete labile nitrogen and carbon sources, the microcosms were preincubated in the dark at 30 °C for 7 days. Following preincubation, rice straw (1 g/100 g mixed soil) was added as a carbon source.

For the nitrate concentration gradient experiment, sodium nitrate was supplemented at concentrations of 0, 2, 10, 30, 60, and 80 μmol/g soil based on the natural nitrate concentration (3.02 μmol/g-dry soil) (Appendix A). To inhibit the reduction of nitrous oxide to nitrogen_,_ the gas phase in the vials was replaced with a mixture of 90% Ar and 10% C_2_H_2_ [26]. For the nitrous oxide gradient experiment, the gas phase in the microcosms was first replaced with Ar, followed by the introduction of nitrous oxide at concentrations of 0, 10, 20, and 30%.

All microcosms were incubated in the dark at 30 °C for 3 days, and each treatment was performed in triplicate. The experimental design for the nitrate and nitrous oxide gradient treatments is shown in Appendix A. Gas samples were collected at 4 h intervals and stored in 15 mL glass serum vials for analysis of nitrogen dynamics. The methods for measuring nitrous oxide and ammonium concentrations are described in the Appendix A.

### 2.3. Soil Microcosms for DNA-SIP Incubation

A randomized controlled experimental design was employed, including three nitrogen source treatments (+^15^NO_3_^−^ or ^14^NO_3_^−^, +^15^N_2_O or ^14^N_2_O, and +^15^N_2_ or ^14^N_2_) and one soil-only control (CK). Each treatment was conducted in triplicate to ensure reproducibility and to allow for statistical analysis.

Microcosms were prepared using the procedure as described in Section 2.2. In total, three experimental treatments with different nitrogen sources (NO_3_, N_2_O, and N_2_) and one soil control treatment were used: (i) soil +Na^15^NO_3_ (+^15^NO_3_^−^) or Na^14^NO_3_ (+^14^NO_3_^−^) (30 μmol/g mixed soil); (ii) soil +^15^N_2_O (+^15^N_2_O) or ^14^N_2_O (+^14^N_2_O) (^15 or 14^N_2_O/Ar = 20:80, *v*/*v*); (iii) soil +^15^N_2_ (+^15^N_2_) or ^14^N_2_ (+^14^N_2_) (^15 or 14^N_2_/Ar = 80:20, *v*/*v*); and (iv) a soil-only control (CK). Pure ^15^N_2_, ^15^N_2_O (>99.9%, GL Sciences, Inc., Tokyo, Japan), and ^15^NaNO_3_ (>99.8%, SI Science, Saitama, Japan) were used in this study. The vials were filled with Ar for treatments (i) and (iv). All microcosms were incubated in the dark at 30 °C for 24, 48, or 72 h. Gas samples from all four treatments were collected and stored in 15 mL glass serum vials for nitrogen dynamics analysis. Soil samples from the three experimental treatments were destructively collected for DNA-SIP analysis.

### 2.4. Determination of ^15^N-Labeled Gas

To quantify the consumption or generation of nitrous oxide and nitrogen gas in the microcosms containing nitrate, nitrous oxide, or nitrogen as substrates, headspace gas from each vial was collected using a syringe and stored in 15 mL glass serum vials. Prior to analysis, the gas samples were diluted with ultrapure helium (He). Gas chromatography-mass spectrometry (GC-MS) was performed using a GC-MS system (GC-2014 gas chromatograph and HS-20 headspace sampler, Shimadzu, Kyoto, Japan) equipped with an SH-Rt-Q-BOND column (32.5 m, 0.32 mm i.d., 10 µm film thickness, Shimadzu, Kyoto, Japan). Ultrapure helium was used as the carrier gas at a flow rate of 2 mL/min. A 1 mL gas sample was injected via the sample loop into the separation column, which was heated at 50 °C, with a split ratio of 30. For MS detection following electron-impact ionization, the detection voltage was set to 0.8 kV, the injector temperature was 250 °C, and the ion-source temperature was 200 °C. Mass spectra were obtained in the selected ion monitoring mode. Other details regarding gas sample injection, GC separation, and MS detection were consistent with a previous study [27]. Pure ^15^N_2_ and ^15^N_2_O were diluted in ultrapure He (1–2 × 10^4^ ppm) to generate standard curves for the GC-MS analysis.

### 2.5. DNA Extraction, SIP Gradient Fractionation, and Quantitative PCR

Soil DNA and RNA were extracted from 0.5 g of soil collected from each microcosm in the three experimental treatments on days 0–3, following a previously described method [25]. The collected nucleic acid solution was purified to remove the PCR inhibitors using a OneStep PCR Inhibitor Removal Kit (Zymo Research, Orange, CA, USA) according to the manufacturer’s protocol. RNA was removed using RNase A (Thermo Fisher Scientific, Waltham, MA, USA) following the manufacturer’s instructions. The resulting RNA-free DNA solution was purified using a QIAquick PCR Purification Kit (Qiagen, Venlo, The Netherlands) and quantified using a NanoDrop One (Thermo Fisher Scientific).

Isopycnic gradient centrifugation and fractionation were performed as previously described using a saturated CsCl solution with a buoyant density of 1.88 g/mL [28,29]. A total of 2.5 μg of extracted DNA was dissolved in a mixed solution of CsCl and gradient buffer (density: 1.690 g/mL) and transferred to a 5PA seal tube (Eppendorf Himac Technologies Co., Ltd., Hitachinaka, Ibaraki, Japan). The DNA mixture was ultracentrifuged at 20 °C and 55,000 rpm (RCF, 172,750× *g*) for 66 h using a Himac Micro Ultracentrifuge CS100 FNX (Eppendorf Himac Technologies Co., Ltd., Japan) equipped with a Himac S110AT-2611 rotor (Eppendorf Himac Technologies Co., Ltd.; k-factor = 15). To improve the resolution and ensure adequate separation of labeled and unlabeled DNA, each seal tube was immediately fractionated into 25 equal fractions using a model SPDC-1 syringe pump (Asone, Osaka, Japan) after ultracentrifugation. The buoyant density (BD) of each fraction was measured using a model PR-RI digital refractometer (ATAGO, Co., Ltd., Tokyo, Japan). The DNA recovered from each fraction was precipitated using a polyethylene glycol solution (30% PEG, 1.6 M NaCl) with 20 μg glycogen and washed with 70% ethanol, then resuspended in 30 μL TE buffer (pH 8.0).

Quantitative PCR (qPCR) was performed in triplicate using the 27F/520R primer set, TB Green Premix Ex Taq (TaKaRa Bio, Shiga, Japan) and the StepOnePlus System (Applied Biosystems, Foster City, CA, USA) to quantify the 16S rRNA gene copy number in each collected fraction. Standard curves were generated using a 10-fold serial dilution of standard DNA (8.883 × 10^5^ to 8.883 × 10^11^). The 16S rRNA gene copy numbers were calculated from the Ct values using the standard curve equation. The PCR conditions and the standard curve for each target gene were consistent with those described previously [25].

### 2.6. 16S rRNA Gene Amplicon Sequencing

On the basis of the qPCR results, DNA samples collected from the heavy and light gradient fractions in each selected treatment were prepared for 16S rRNA gene (V3–V4) amplicon sequencing to analyze the bacterial community composition using primers 341F (5′-CCTACGGGNGGCWGCAG-3′) and 805R (5′-GACTACHVGGGTATCTAATCC-3′). Fractions were selected based on evident contrasts between the labeled and unlabeled samples at the same buoyant DNA densities. The selected treatments and corresponding fractions were as follows: (i) ^15^NO_3_^−^ treatment (heavy fractions H0, H1, and H2) or ^14^NO_3_^−^ treatment (light fractions L1, L2, and L3) at 24 h of culture; (ii) ^15^N_2_O treatment (heavy fractions H1 and H2) or ^14^N_2_O treatment (light fractions L2 and L3) at 48 h of culture; and (iii) ^15^N_2_ treatment (heavy fractions H0, H1, and H3) or ^14^N_2_O treatment (light fractions L1, L2, and L3) at 72 h of culture. Each fraction was analyzed in triplicate. The details of the 16S rRNA gene amplicon sequencing method are described in Appendix A. Raw 16S rRNA gene sequences were generated using the Illumina MiSeq platform (2 × 300 bp paired-end). Reads whose 5′ ends exactly matched the primer sequences were extracted using the FASTX-Toolkit (ver. 0.0.14), and the primer sequences were removed. Low-quality reads with a quality score < 20 were discarded using Sickle (ver. 1.33). Paired-end reads were then merged using FLASH (ver. 1.2.11) with the following parameters: merged read length = 310 bp, read length = 230 bp, and overlap length = 10 bp. After removing chimeric and noisy sequences using the DADA2 plugin in QIIME 2 (ver. 2021.2), representative sequences and an ASV table were generated. Taxonomic classification was performed using the feature-classifier plugin by comparing the representative sequences with 99% of the operational taxonomic unit (OTU) reference sequences from the SILVA database (ver. 138.1; Appendix A [30]).

### 2.7. RNA Extraction and Metatranscriptomic Analysis

Samples from the +^15^NO_3_^−^ and +^15^N_2_O treatments incubated for 24 and 48 h, respectively, were selected for metatranscriptome sequencing. Each treatment was analyzed in triplicate. DNA and RNA were extracted from the soil (0.5 g) using the protocol described in Section 2.5, including purification with the OneStep PCR Inhibitor Removal Kit (Zymo Research, Orange, CA, USA). Genomic DNA was removed from the extracted DNA and RNA solutions (the same as those used for SIP analysis) using DNase I (Nippon Gene, Tokyo, Japan), following the manufacturer’s instructions. The GeneJET RNA Cleanup and Concentration Kit (Thermo Fisher Scientific, Waltham, MA, USA) was used to remove DNA degradation products and concentrate RNA. The quality, concentration, and integrity of the extracted RNA were assessed using a NanoDrop™ 1000 spectrophotometer (Thermo Fisher Scientific), agarose gel electrophoresis, and a model 2100 bioanalyzer instrument (Agilent 2100, Agilent Technologies, Santa Clara, CA, USA).

High-quality RNA samples (optical density [OD] 260/280 = 1.8–2.2; OD260/230 ≥ 2.0, RNA Integrity Number [RIN] ≥ 6.5, 28S:18S ≥ 1.0) were used for library construction. Detailed concentrations and volumes are provided in Appendix A. RNA libraries were prepared using the MGIEasy Fast RNA Library Prep Set, according to the manufacturer’s instructions (MGI Tech, Shenzhen, China). Library quantification was performed using a Synergy H1 microplate reader (Agilent Technologies, CA, USA) in combination with a QuantiFluor dsDNA System (Promega, Madison, WI, USA). Metatranscriptomic sequencing was performed using the DNBSEQ platform with a 150 bp paired-end configuration at the Bioengineering Laboratory Co. (Kanagawa, Japan). Quality metrics of the raw metatranscriptomic reads are summarized in Appendix A. Raw paired-end reads were subjected to quality-trimmed using Trimmomatic v0.36 (https://github.com/usadellab/Trimmomatic, (accessed on 10 May 2024)) with the following parameters: SLIDINGWINDOW/4:15 MINLEN/75 to remove adaptor contaminants and low-quality reads. The clean reads were then aligned to the SILVA SSU (16S/18S) and SILVA LSU (23S/28S) databases using SortMeRNA v2.1b (https://github.com/sortmerna/sortmerna, (accessed on 10 May 2024)) software to remove rRNA-related reads. Clean data were assembled using MEGAHIT v1.1.1-2-g02102e1 (https://github.com/voutcn/megahit, (accessed on 10 May 2024)). Genes were predicted using METAProdigal (https://github.com/hyattpd/Prodigal, (accessed on 10 May 2024)). A nonredundant gene catalog was constructed with 95% identity and 90% coverage using CD-HIT (https://github.com/weizhongli/cdhit/releases [31], (accessed on 10 May 2024)). Open reading frames (ORFs) of the assembled contigs were predicted using Prodigal (v2.6.3), and all ORFs were clustered into unique genes using CD-HIT V4.8.1 (https://github.com/weizhongli/cdhit/releases (accessed on 15 May 2024)) [31]. The unique transcriptome gene set was annotated using the Kyoto Encyclopedia of Genes and Genomes (KEGG) database with Kofam v1.2.0 [32]. Functional protein sequences were compared using NCBI Protein BLAST (https://blast.ncbi.nlm.nih.gov/Blast.cgi?PROGRAM=blastp (accessed on 10 June 2024)) (Basic Local Alignment Search Tool) to identify the microorganisms responsible for nitrogen transformation. Transcript abundance was normalized to transcripts per million (TPM) using SAMtools version 1.18.

### 2.8. Statistical Analyses

The relative abundances of the 16S rRNA gene sequences in the +^15^N and +^14^N treatment fractions was calculated by normalizing the number of amplicon sequence reads to the copy numbers of the 16S rRNA genes obtained using qPCR. Linear discriminant analysis (LDA) effect size (LEfSe) was used to identify taxa that were significantly more abundant in the +^15^N treatments than in the +^14^N treatments [33]. A nonparametric factorial Kruskal–Wallis sum-rank test (*p* < 0.05) was performed to detect significantly different features. Bioinformatics analyses, including LEfSe V2.1 and bubble diagram visualization V1.0, were performed using OmicStudio tools (https://www.omicstudio.cn/tool/) [34]. Based on the KEGG annotations, the key nitrogen transformation pathways and their corresponding functional genes are summarized in Appendix A.

## 3. Results

### 3.1. Determination of SIP Experimental Conditions

Our previous study demonstrated that an incubation period of approximately 72 h under 80% ^15^N_2_ is suitable for SIP studies targeting nitrogen fixation [35]. To ensure consistent reaction times across experimental groups during DNA-SIP while balancing labeling efficiency, microbial activity, and environmental relevance, we optimized the incubation duration and the substrate concentrations of nitrate (2, 10, 30, 60, and 80 μmol NaNO_3_^−^/g soil) and nitrous oxide (10, 20, and 30%).

In the nitrate concentration gradient experiment, N_2_O generation in the +2 μmol NaNO_3_/g soil treatment ceased within 8 h, contributing to <1% of the total gas content owing to rapid nitrate depletion (Appendix A), which was insufficient for effective ^15^N labeling of microbial DNA in the DNA-SIP experiments. In contrast, in the +10 μmol NaNO_3_/g soil treatment, N_2_O generation ceased within 24 h, accounting for approximately 5% of the total gas content. This coincided with near-complete nitrate depletion and a high conversion rate of nitrate to ammonium (Appendix A). In the +30, +60, and +80 μmol NaNO_3_/g soil treatments, N_2_O was continuously generated for up to 48 h, stabilizing at 12–14% of the total gas content (Appendix A). The +30 μmol treatment maintained nitrate availability for over 48 h, albeit with a marginally lower ammonium conversion efficiency compared with the +10 μmol treatment (Appendix A). In the +60 and +80 μmol treatments, nitrate remained even after 96 h of incubation, indicating that excessive nitrate levels significantly deviated from natural paddy field conditions (Appendix A).

In the nitrate gradient experiment, the maximum N_2_O concentration generated was approximately 12–14%. To match the maximum N_2_O concentration generated in the nitrate experiment (Appendix A) and ensure the continuous availability of N_2_O during the incubation period, a concentration range of 10–30% N_2_O was selected for the N_2_O concentration gradient experiment. As shown in Appendix A, N_2_O was fully depleted after 48 h in the +10% N_2_O treatment, whereas in the +20% N_2_O treatment, depletion was nearly complete by 72 h. In contrast, the +30% N_2_O treatment resulted in excess residual N_2_O.

To maintain consistency with the previous 80% ^15^N_2_ conditions and the newly optimized +NO_3_^−^ and +N_2_O conditions, the DNA-SIP incubation times were set to 0, 24, 48, and 72 h. A nitrate concentration of 30 μmol/g was selected, as this value sustained the reaction for over 48 h while maintaining ammonium conversion efficiency. Additionally, 20% N_2_O was chosen as the optimal condition for the 72 h incubation. These conditions ensured effective ^15^N labeling while preventing excessive substrate accumulation.

### 3.2. Inorganic Nitrogen Reduction and ^15^N-Labeled Gas Generation

In the +^15^NO_3_ treatment, ^15^N_2_O accumulated at 24 h (Figure 1a), indicating that denitrification reactions occurred before this time. Additionally, ^15^N_2_ accumulation was observed at 72 h, suggesting that the generated ^15^N_2_O was converted to ^15^N_2_ between 48 and 72 h (Figure 1a).

In the +^15^N_2_O treatment, ^15^N_2_O was consumed and a portion of it was converted to ^15^N_2_ between 24 and 48 h (Figure 1b). Interestingly, the total amount of ^15^N_2_O consumed exceeded that of ^15^N_2_ generated, suggesting that either the generated ^15^N_2_ was partially fixed by microorganisms or not all of the added ^15^N_2_O was fully converted to ^15^N_2_.

In the +^15^N_2_ treatment, the concentration of ^15^N_2_ decreased during the incubation period, indicating that nitrogen fixation had occurred (Figure 1c).

No detectable gas production was observed in the blank control group during the incubation.

### 3.3. Quantification of the 16S rRNA Gene in CsCl Gradient Fractions

The abundance of the 16S rRNA gene across the CsCl gradient fractions was compared using qPCR to assess the bacterial incorporation of ^15^N under different nitrogen source treatments. According to the qPCR results, the maximum relative abundance of the 16S rRNA genes significantly shifted to heavier fractions (BD: 1.6729–1.6801 g/mL) compared to lighter fractions (BD: 1.6678–1.6751 g/mL) in all three treatments (Figure 2), indicating that the DNA of the cultured bacteria had successfully incorporated ^15^N. The absolute 16S rRNA gene copy numbers across the CsCl gradient fractions for each treatment, as determined by qPCR, are shown in Appendix A.

Interestingly, in the +^15^N_2_ treatment, 16S rRNA gene abundance shifted to a heavier fraction owing to N_2_ fixation after 72 h of culture, whereas in the +^15^N_2_O treatment, the shift occurred as early as 48 h. These results suggest that ^15^N assimilation occurred at a faster rate in the +^15^N_2_O treatment (N_2_O → N_2_ → NH_4_^+^) than in the +^15^N_2_ treatment (N_2_ → NH_4_^+^).

### 3.4. Potential ^15^N-Assimilating Microorganisms Revealed by DNA-SIP and 16S rRNA Amplicon Sequencing

To identify the microorganisms assimilating ^15^N, samples were selected at different time points: after 72 h for the +^15^N_2_ treatment, after 48 h for the +^15^N_2_O treatment, and after 24 h for the +^15^NO_3_ treatment. 16S rRNA amplicon sequencing was performed on the CsCl gradient fractions containing the highest concentrations of 16S rRNA genes in the heavy fractions (H0–H2) and the corresponding BD of the light fractions (L1–L3) (Figure 2). LEfSe analysis was used to identify microorganisms that exhibited significant increases after ^15^N labeling compared with the unlabeled treatments within the same BD. An LDA score threshold was applied to identify taxa with biologically meaningful differences, enabling the robust detection of key microbial taxa actively assimilating ^15^N under different nitrogen treatments.

In the +^15^NO_3_^−^ treatment, the primary ^15^N-assimilating microorganisms (LDA score ≥ 3.5) were *Gemmatimonadaceae*, *Anaerolineaceae*, *Bacillus*, *Geomonas*, *Xanthomonadaceae*, *Arenimonas*, and *Clostridium* (Figure 3a). In the +^15^N_2_O treatment, the dominant ^15^N-assimilating bacteria (LDA score > 4.3) were *Sulfurospirillum*, *Geomonas*, and *Oryzomicrobium* (Figure 3b). In the +^15^N_2_ treatment, the major N-assimilating microorganisms (LDA score > 4.2) were *Clostridium*, *Geomonas*, and *Pseudomonadaceae*. In addition, *Bacillus*, *Oryzomicrobium*, and other bacteria exhibited significant ^15^N assimilation (LDA score > 3.7) (Figure 3c).

Among the ^15^N-assimilating bacteria identified at the genus level, *Geomonas* was the only taxon detected in all treatments. *Oryzomicrobium* was involved in ^15^N assimilation in the +N_2_O and +N_2_ treatments, whereas *Clostridium* and *Bacillus* were active in the +^15^NO_3_ and +N_2_ treatments. These bacteria demonstrate a strong capacity for nitrogen incorporation from the environment, with *Geomonas*, *Clostridium*, and *Bacillus* exhibiting a preference for nitrogen assimilation in both liquid and gaseous forms.

### 3.5. Functional Gene Transcripts and Bacteria Involved in Ammonium Generation Revealed by Metatranscriptomics

In the +^15^NO_3_^−^ treatment, expressions of functional genes related to denitrification (*narG*/*napA*, *nirK*/*S*, *norB*, and *nosZ*) were most pronounced (Figure 4b(left)). Among these genes, *narG*/*napA*, which catalyzes the reduction of NO_3_^−^ to NO_2_^−^, was predominantly transcribed by *Bacillaceae* (38.4%) and *Paenibacillaceae* (27.9%). The expression of *nirK*/*S* (NO_2_^−^ → NO) was attributed primarily to *Anaerolineaceae* (14.1%), *Ardenticatenaceae* (11.9%), and *Bacillaceae* (5.6%). Similarly, *norB* (NO → N_2_O) was transcribed mainly by *Bacillaceae* (51.3%), *Anaeromyxobacteraceae* (13.1%), *Comamonadaceae* (3.3%), and *Holophagaceae* (2.9%), while *nosZ* (N_2_O → N_2_) was expressed predominantly by *Bacillaceae* (79.4%) (Figure 4c(left)). Although the expression levels of genes related to nitrogen fixation and DNRA were low in this treatment group (Figure 4b(left)), *nrfA*, a gene involved in DNRA, was transcribed by *Bacillaceae* (44.4%), *Anaeromyxobacteraceae* (10.3%), and *Holophagaceae* (6.7%). Moreover, nitrogenase genes (*nifD*/*K*) were expressed by *Syntrophobacteraceae* (77.3%) and *Clostridiaceae* (9.2%). In contrast, the genes associated with nitrate and nitrite assimilation were more actively transcribed (Figure 4b(left)). *Clostridiaceae* and *Bacillaceae* were implicated in nitrite assimilation (*NIT-6*/*nirA*/*NasBDE*), whereas *Bacillaceae* was involved in nitrate assimilation (Figure 4c(left)). These findings suggest that, under the +^15^NO_3_^−^ treatment, denitrification was the predominant nitrogen transformation process, followed by nitrite and nitrate assimilation. In contrast, nitrogen fixation and ammonium generation via the DNRA were less active. Notably, *Bacillaceae* dominated the transcription of genes related to denitrification, DNRA, and nitrate or nitrite assimilation, indicating high metabolic activity under nitrate-rich conditions.

In the +^15^N_2_O treatment, genes related to nitrogen fixation (*nifD*/*K*) showed the highest expression and were primarily transcribed by *Rhodocyclaceae* and *Sulfurospirillaceae* (82.4% and 10.7%, respectively) (Figure 4c(right)). The second most highly expressed gene was nitrous oxide reductase gene (*nosZ*), with *Sulfurospirillaceae*, *Comamonadaceae*, and *Rhodocyclaceae* being the primary contributors to its transcription (51.8%, 35.2%, and 7.1%, respectively). Although the expression level of the DNRA-related gene *nrfA* was relatively low in this study (Figure 4b(right)), it was predominantly transcribed by *Anaeromyxobacteraceae* (15.1%), *Sulfurospirillaceae* (6.6%), and *Geobacteraceae* (2.7%). Additionally, the ammonium assimilation gene (*amt*) was expressed in *Rhodocyclaceae* (33.7%), *Sulfurospirillaceae* (12.3%), *Anaeromyxobacteraceae* (10.3%), *Geobacteraceae* (5.2%), and *Clostridiaceae* (4.1%). A minor expression of genes involved in the early steps of denitrification (NO_3_^−^ → NO_2_^−^ → NO) was also detected, suggesting the transient formation of nitrate or nitrite (Figure 4b(right)). However, the lack of expression of *hao* and *amoA* expression, which are associated with nitrification, indicated that neither nitrate nor nitrite were produced via this pathway (Figure 4b(right)). These findings indicate that, under the +^15^N_2_O treatment, the primary nitrogen transformation processes were N_2_O reduction to N_2_ and nitrogen fixation (N_2_O → N_2_ → NH_4_^+^), followed by ammonium assimilation. Furthermore, under these experimental conditions, the conversion of small amounts of nitrite into ammonium was considered to be driven by bacteria such as *Anaeromyxobacteraceae*, although the source of nitrite remains unidentified.

## 4. Discussion

The combined DNA-SIP and metatranscriptomics analyses provided a comprehensive view of inorganic nitrogen transformation in soil under three treatments: +^15^N_2_, +^15^N_2_O, and +^15^NO_3_^−^. Although DNA-SIP alone may be influenced by potential cross-feeding, its integration with metatranscriptomics helps mitigate this limitation. These analyses reveal the complexity of microbial processes and their critical roles in the nitrogen cycle, particularly in ammonium generation.

In the +^15^NO_3_^−^ treatment, metatranscriptomics revealed that denitrification (NO_3_^−^ → NO_2_^−^ → NO → N_2_O → N_2_) and nitrate or nitrite assimilation were the most active nitrogen-related processes. Ammonium generation appeared to be less prominent, likely because of the low carbon-to-nitrogen (C/N) ratio set for the experiment, which typically suppresses DNRA [36]. However, ammonium accumulation was observed in the soil after prolonged incubation (Appendix A), suggesting that even if microorganisms directly assimilated NO_3_^−^ or fixed N_2_, ammonium could be retained in the soil following microbial cell death. In terms of nitrogen assimilation, SIP analysis identified *Geomonas* belonging to *Geobacteraceae*, *Anaerolineaceae*, *Gemmatimonadaceae*, *Bacillaceae*, *Anaeromyxobacteraceae*, and *Clostridiaceae* as the primary taxa incorporating ^15^N (Figure 3a). To determine whether these bacteria directly assimilated nitrate or produced ammonium before uptake, a metatranscriptomic analysis was performed. *Bacillaceae* not only directly assimilated nitrate or nitrite (*nrt*/*nas*) but also generated ammonium via DNRA (*nrfA*), and subsequently incorporated ammonium through ammonium transporters (*amt*) (Figure 4c(left)). In contrast, *Anaeromyxobacteraceae*, despite lacking *amt* expression, assimilated nitrate (*nrt*), incorporated ^15^N into DNA, and generated ammonium via DNRA (*nrfA*). *Clostridiaceae* incorporated ^15^N into DNA through nitrite assimilation (*nas*) and nitrogen fixation (*nifD*/*K*). Interestingly, *Geomonas*, *Anaerolineaceae*, and *Gemmatimonadaceae*, which were identified as key ^15^N-assimilating bacteria in the SIP analysis, did not express nitrogen-related functional genes such as *nrt/nas*, *amt*, or *nifD/K* (Figure 4c(left)). Although *Geomonas* is a highly active diazotroph in paddy soils [35,37], *nifD/K* expression was not detected in the present study. Conversely, *Syntrophaceae* and *Syntrophobacteraceae*, which transcribed nitrogen fixation genes (*nifD/K*), were not identified as ^15^N-assimilating bacteria in the SIP analysis. These discrepancies may reflect a time lag between RNA expression and DNA labeling [38,39,40]. *Geomonas*, *Anaerolineaceae*, and *Gemmatimonadaceae* likely exhibited high RNA expression levels before the SIP detection period (24 h), whereas *Syntrophaceae* and *Syntrophobacteraceae* may have incorporated ^15^N into their DNA after metatranscriptomic sampling. These findings suggest that although denitrification dominated the +^15^NO_3_^−^ treatment, ammonium generation via DNRA or denitrification and nitrogen fixation was primarily carried out by *Bacillaceae*, *Anaeromyxobacteraceae*, *Clostridiaceae*, and *Geomonas*.

In the +^15^N_2_O treatment, the metatranscriptomic analysis revealed N_2_O reduction and nitrogen fixation (N_2_O → N_2_ → NH_4_^+^) as the dominant processes (Figure 4b(right)). These reactions were driven primarily by *Rhodocyclaceae* (including *Oryzomicrobium*) and *Sulfurospirillaceae* (including *Sulfurospirillum*), which were also identified as the dominant taxa of ^15^N-assimilating bacteria based on SIP analysis (Figure 3b and Figure 4c(right)). Additionally, *Geobacteraceae*, which showed high ^15^N assimilation in the SIP analysis, exhibited extracellular reduction of nitrate to ammonium (via *narG*/*napA* and *nrfA*), followed by ammonium uptake (via *amt*), reinforcing the consistency between the SIP results and metatranscriptomic findings (Figure 3b). Thus, in the +^15^N_2_O treatment, *Rhodocyclaceae*, *Sulfurospirillaceae*, and *Geobacteraceae* were suggested to generate ammonium via nitrogen fixation and DNRA. The occurrence of DNRA indicated the presence of nitrate in the soil. Interestingly, however, no functional genes associated with nitrate or nitrite production (e.g., *hao* and *amo*) were detected in the metatranscriptomic analysis. Additionally, the SIP results showed that ^15^N assimilation occurred more rapidly in the +^15^N_2_O treatment than in the +^15^N_2_ treatment (Figure 2). Moreover, the consumption of ^15^N_2_O exceeded the production of ^15^N_2_ based on GC-MS analysis (Figure 1a,b). These findings suggest that N_2_O may be consumed through pathways other than its reduction to N_2_ followed by fixation as NH_4_^+^. Frutos et al. proposed that the direct conversion of N_2_O into nitrate or nitrite is an unknown reaction with thermodynamic advantages [41]. The present findings indicate that it is plausible that if such a pathway exists, the resulting nitrate or nitrite may have been further reduced to ammonium and assimilated into the microbial biomass.

In the +^15^N_2_ treatment, the SIP analysis revealed *Clostridium* belonging to *Clostridiaceae*, *Geomonas* belonging to *Geobacteraceae*, and *Pseudomonadaceae* (including *Pseudomonas* and *Azotobacter*) were the primary N-fixing bacteria that generated intracellular ammonium (Figure 3c). *Geomonas* [35,37], along with *Clostridium* and *Pseudomonadaceae* [42,43], have been reported as active diazotrophs in paddy soils.

A notable difference in active nitrogen-fixing bacteria was observed between the +^15^N_2_O and +^15^N_2_ treatments (Figure 3b,c and Figure 4c(right)). In the +^15^N_2_O treatment, *Oryzomicrobium* and *Sulfurospirillum* were the dominant nitrogen-fixing bacteria, whereas in the +^15^N_2_ treatment, *Clostridium*, *Geomonas*, and *Pseudomonadaceae* (including *Pseudomonas* and *Azotobacter*) were dominant. This difference may be attributed to the absence of the *nosZ* gene in *Geomonas*, *Clostridium*, and *Azotobacter*, which prevents members of these taxa from reducing N_2_O to N_2_. Consequently, *Rhodocyclaceae* (including *Oryzomicrobium*) and *Sulfurospirillaceae* (including *Sulfurospirillum*), which possess *nosZ* and efficiently utilize N_2_O for nitrogen fixation, likely outcompete *Clostridium*, *Geomonas*, and *Pseudomonadaceae*.

Overall, this study reveals that denitrifiers, such as *Rhodocyclaceae* and *Bacillaceae*, primarily converted nitrate into nitrogen gas; diazotrophs, including *Geobacteraceae*, *Bacillaceae*, *Pseudomonadaceae*, *Rhodocyclaceae*, *Sulfurospirillaceae*, and *Clostridiaceae*, converted nitrogen gas into ammonium; and DNRA bacteria such as *Sulfurospirillaceae* and *Anaeromyxobacteraceae* mainly competed with denitrifiers for nitrate or nitrite and reduced these compounds to ammonium. Furthermore, bacterial families, such as *Geobacteraceae* (under +N_2_O and +N_2_ conditions), *Bacillaceae* (under +NO_3_^−^ and +N_2_ conditions), *Rhodocyclaceae* (under +N_2_O and +N_2_ conditions), *Anaeromyxobacteraceae* (under +NO_3_^−^ and +N_2_O conditions), and *Clostridiaceae* (under +NO_3_^−^ and +N_2_ conditions), actively contributed to ammonium generation under multiple nitrogen substrate conditions in the paddy soil environment (Figure 5).

Compared with other soils, paddy soils are frequently exposed to anaerobic conditions owing to prolonged flooding and are rich in iron, sulfur, and organic carbon derived from rice plants. These factors provide electron donors and acceptors that support dominant microbial communities [44,45]. This study also confirmed that bacteria already known as Fe(III)-reducing diazotrophs, which include *Geobacteraceae*, *Anaeromyxobacteraceae*, *Clostridiaceae*, and *Pseudomonadaceae*, generated ammonium through nitrogen fixation [44,46,47,48]. Furthermore, iron has been reported to contribute to ammonium generation through nitrate-dependent iron oxidation (NRFO) and Fe(II)-dependent DNRA (nitrate respiration-associated denitrification), confirming that *Anaeromyxobacteraceae* play a key role in denitrification, DNRA, and iron reduction [49]. A previous study revealed a strong correlation between DNRA and sulfate reduction rates [50]. This study further demonstrated that *Sulfurospirillaceae*, a family capable of sulfate reduction, are involved in DNRA. Additionally, organic carbon derived from rice plants, such as rice straw or other residues, is actively decomposed and utilized by *Clostridiaceae* [51]. The decomposition products generated in this process are subsequently utilized by families including *Geobacteraceae* and *Anaeromyxobacteraceae* [37,52,53]. Furthermore, species within the *Rhodocyclaceae* family, which metabolize recalcitrant organic compounds that include phenolic substances [54], may contribute to lignin degradation from rice straw while simultaneously generating ammonium in paddy soils. Notably, *Rhodocyclaceae* have also been implicated in the regeneration of rice leaves following damage, whereas *Geobacteraceae* and *Clostridiaceae* promote rice recovery from damage or environmental stress, directly supporting rice growth [55].

The findings presented in this study provide a comprehensive overview of the ammonium-generating microbial consortia in paddy soils and suggest that ammonium generation may be closely linked to the reactions of abundant substances such as iron, sulfur, and rice residues in paddy environments. By focusing on the bacterial consortia primarily responsible for ammonium generation and the additional processes they facilitate, it may be possible to enhance microbial activity and consequently promote ammonium generation.

## 5. Conclusions

This study employed ^15^N-DNA-SIP and metatranscriptomics to provide the first comprehensive insights into ammonium-generating microbial consortia, focusing specifically on DNRA, denitrification, and nitrogen fixation in paddy soil microcosms amended with ^15^NO_3_^−^, ^15^N_2_O, and ^15^N_2_ as nitrogen sources. The results indicated that members of the families *Holophagaceae*, *Bacillaceae*, *Sulfurospirillaceae*, *Geobacteraceae*, and *Anaeromyxobacteraceae* play a significant role in generating extracellular ammonium via DNRA. Additionally, *Geobacteraceae*, *Bacillaceae*, *Pseudomonadaceae*, *Rhodocyclaceae*, *Sulfurospirillaceae*, and *Clostridiaceae* are identified as key contributors to intracellular ammonium generation through nitrogen fixation. Notably, several taxa, including *Geobacteraceae* (under +N_2_O and +N_2_ conditions), *Bacillaceae* (under +NO_3_^−^ and +N_2_ conditions), *Rhodocyclaceae* (under +N_2_O and +N_2_ conditions), *Anaeromyxobacteraceae* (under +NO_3_^−^ and +N_2_O conditions), and *Clostridiaceae* (under +NO_3_^−^ and +N_2_ conditions), were active under multiple nitrogen substrate conditions, contributing significantly to ammonium generation in the paddy soil environment. Many of these microorganisms, such as those involved in iron or sulfate reduction and rice straw decomposition, are also involved in essential ecological processes characteristic of paddy soils. The study findings revealed ammonium-generating microbial consortia in paddy soils that contain several key bacterial drivers of multiple reductive nitrogen transformations and suggested their diverse functions in paddy soil metabolism. Overall, this study advances our understanding of nitrogen-transforming microbial consortia in paddy soils and may inform sustainable agricultural practices by supporting microbially mediated ammonium generation, improving soil fertility, and reducing the reliance on chemical fertilizers.

## Figures and Tables

**Figure 1 microorganisms-13-01448-f001:**
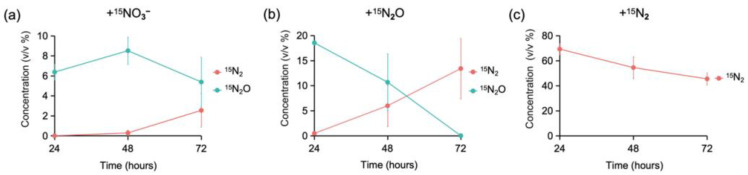
Concentration (*v*/*v* %, volume/volume percent) of ^15^N-labeled gas (^15^N_2_ and ^15^N_2_O, >99.9 atom%) produced and consumed in the microcosm during 1–3 days of incubation across the three treatments: (**a**) ^15^N_2_ generation, ^15^N_2_O generation, and ^15^N_2_O consumption in the +^15^NO_3_^−^ treatment; (**b**) ^15^N_2_O consumption and ^15^N_2_ production in the +^15^N_2_O treatment; (**c**) ^15^N_2_ consumption in the +^15^N_2_ treatment. The *x*-axis shows the incubation time, and the *y*-axis indicates the gas concentration (%). Error bars represent the range of values from three duplicate batch tests.

**Figure 2 microorganisms-13-01448-f002:**
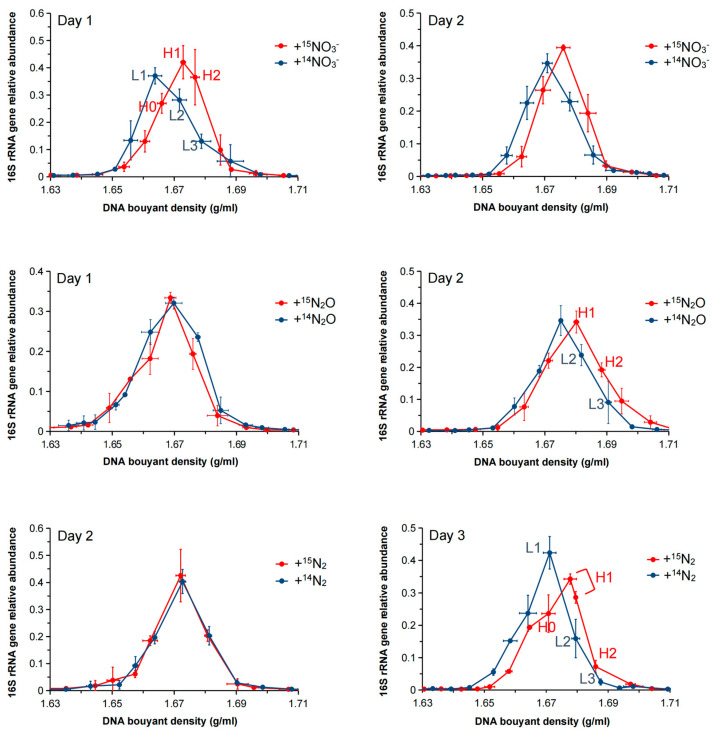
Relative abundance of the 16S rRNA gene across CsCl gradient fractions from the +^15^NO_3_^−^, +^15^N_2_O, and +^15^N_2_ treatments after 1–3 days of incubation. The 16S rRNA gene abundance in each fraction was normalized to the total abundance across all fractions. The *x*-axis indicates buoyant density (g/mL), and the *y*-axis shows the relative abundance of the 16S rRNA gene. Data are presented as the average of triplicate measurements. Vertical error bars represent the standard error of relative abundance. Horizontal error bars represent the standard error of buoyant density for corresponding fractions.

**Figure 3 microorganisms-13-01448-f003:**
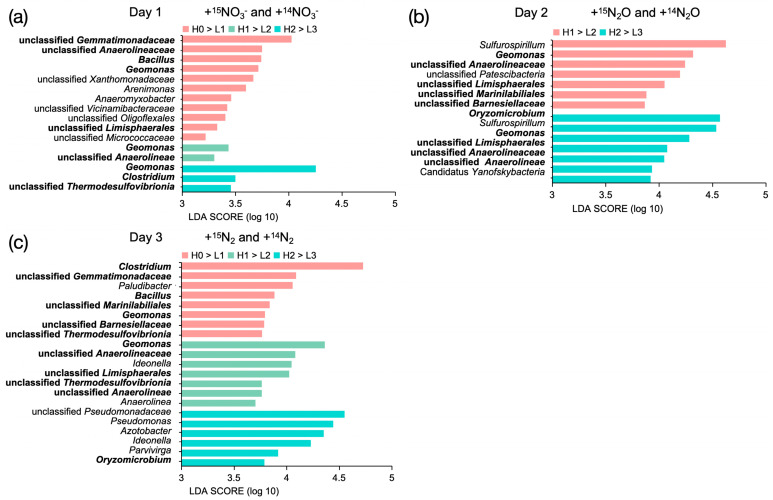
Microorganisms significantly enriched in the ^15^N-labeled treatments compared with the corresponding ^14^N controls within the same CsCl gradient fractions across the three treatments: (**a**) +^15^NO_3_^−^ vs. +^14^NO_3_^−^ after 1 day of incubation; (**b**) +^15^N_2_O vs. +^14^N_2_O after 2 days; and (**c**) +^15^N_2_ vs. +^14^N_2_ after 3 days. Bacterial taxa that incorporated ^15^N across multiple treatments are indicated in bold. The horizontal coordinates indicate the LDA scores calculated for the ^15^N treatment relative to the ^14^N treatment at corresponding density fractions. Data are presented as the average of triplicate measurements. Significance was assessed by a Kruskal–Wallis test (*p* < 0.05) followed by a Wilcoxon test (*p* < 0.05).

**Figure 4 microorganisms-13-01448-f004:**
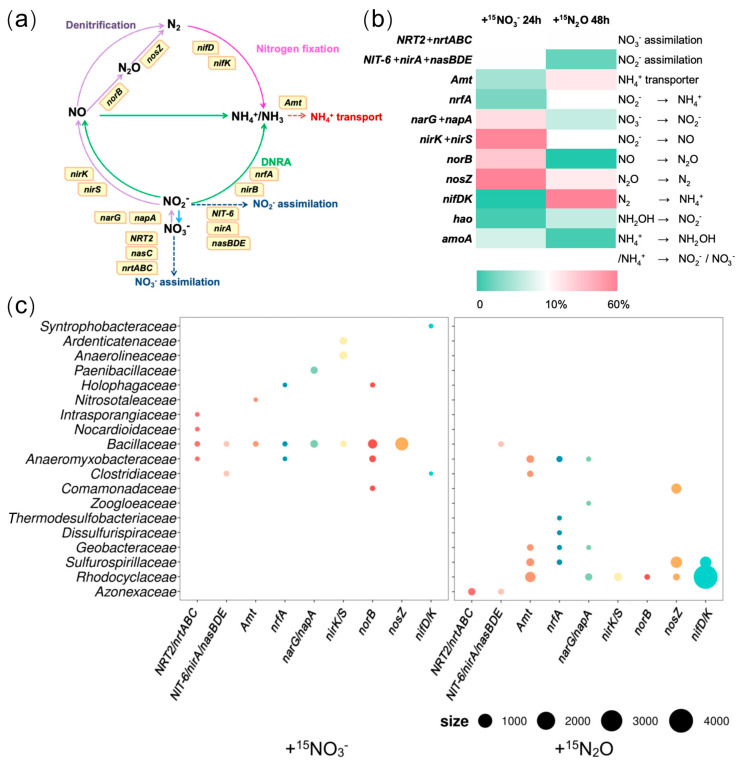
(**a**) Nitrogen cycle reactions and related functional genes. (**b**) Relative expression levels of nitrogen transformation-related genes in the +^15^N_2_O and +^15^NO_3_^−^ treatments, shown as the proportions TPM within each treatment. (**c**) Bacterial taxa expressing nitrogen transformation-related functional genes and their expression levels (based on TPM values), size: TPM. Data are presented as the average of triplicate measurements. Arrow colors indicate each nitrogen transformation process (purple: denitrification; green: DNRA; pink: nitrogen fixation).

**Figure 5 microorganisms-13-01448-f005:**
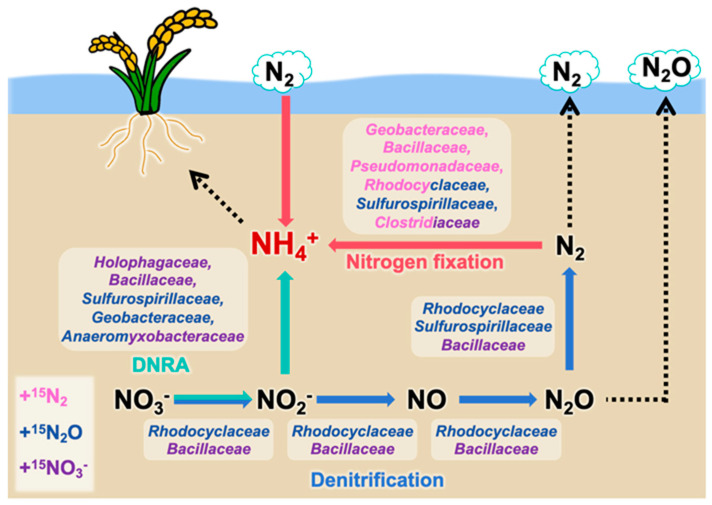
Overview of ammonium-generating microbial consortia in paddy soils, as proposed by the results of this study. Family names in different colors represent nitrogen-metabolizing microbes identified in different nitrogen source addition groups (purple: +^15^NO_3_^−^ treatment; blue: +^15^N_2_O treatment; pink: +^15^N_2_ treatment). Arrow colors indicate each nitrogen transformation process (blue: denitrification; green: DNRA; red: nitrogen fixation).

## Data Availability

The nucleotide sequences obtained by amplicon sequencing and metatranscriptomics in this study have been deposited in the National Center for Biotechnology Information (https://www.ncbi.nlm.nih.gov/) (Bioproject: PRJNA1250227, PRJNA1250229, PRJNA1250230, PRJNA1250231, PRJNA1250232 and PRJNA1250233, and Bioproject: PRJNA1251509 and PRJNA1251506).

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
