# Peer review of "Ammonium-Generating Microbial Consortia in Paddy Soil Revealed by DNA-Stable Isotope Probing and Metatranscriptomics"

_microorganisms, 2025, doi:10.3390/microorganisms13071448_

Round 1
Reviewer 1 Report
Comments and Suggestions for Authors
I found your methods are clear and the experimental design logical. The authors selected important soil conditions and optimized incubation times. The results show how different microorganisms respond to nitrate, nitrous oxide, and nitrogen gas.
I suggest the following small improvements:
Please fix the reference error in the methods ..........Error! Reference source not found.
It would be helpful to add a table summarizing the genes found in the RNA analysis and what nitrogen process they are involved in.
For the LEfSe results, please include more details about which taxa were statistically different and how strong the changes were.
Include more discussion about the control treatment (CK) and how it compares to the others. Although cold storage can help, we know that microorganisms continue to process materials and metabolism is not a direct process, but there are many transformations of biomass, including that of the microorganisms themselves.
Please add one or two sentences explaining the limitations of SIP, especially that some microbes may use ¹⁵N from others (cross-feeding).
Author Response
We sincerely thank Reviewer 1 for the positive and thorough reviews. To address the reviewer’s comments, we have revised the manuscript. We believe that this revision has made our manuscript more comprehensive and solid.
・Please fix the reference error in the methods ..........Error! Reference source not found.
→Thank you for your comment. We have modified the references (Lines 153 and 432)
・It would be helpful to add a table summarizing the genes found in the RNA analysis and what nitrogen process they are involved in.
→Thank you for your comment. Table S6 has been added to the revised manuscript.
・For the LEfSe results, please include more details about which taxa were statistically different and how strong the changes were.
→Thank you for your comment. In the revised manuscript, we have included detailed information on the taxa showing significant differences as identified by the LEfSe analysis, along with their corresponding LDA scores. These scores reflect the magnitude of the differences and highlight key microbial taxa that actively assimilate 15N under different nitrogen treatments. The relevant taxa and their LDA scores are presented in Figure 3 and described in the Results section.
・Include more discussion about the control treatment (CK) and how it compares to the others. Although cold storage can help, we know that microorganisms continue to process materials and metabolism is not a direct process, but there are many transformations of biomass, including that of the microorganisms themselves.
→Thank you for your valuable comments. As our experimental design focused on comparing various 15N-labeled treatments, with 14N serving as the primary control, the original manuscript did not include a detailed discussion of the blank control group. In the actual experiment, we processed the samples as quickly as possible after collection and immediately initiated incubation, thereby minimizing changes in sample metabolism and microbial activity to ensure the reliability of the results. Additionally, GC-MS analysis revealed no significant microbial metabolic changes in the blank control group, which may be attributed to the depletion of available nutrients in the soil during the pre-incubation period. A brief explanation of the control group will be appropriately included in the revised manuscript (Lines 306–307).
・Please add one or two sentences explaining the limitations of SIP, especially that some microbes may use ¹⁵N from others (cross-feeding).
→Thank you for your comment. We have modified the description (Lines 424-425).
Reviewer 2 Report
Comments and Suggestions for Authors
Introduction
Line 63: Explain how the presence of pseudo-NifH genes led to misinterpretations in terms of understanding the nitrogen cycle or the interplay between DNRA, denitrification and nitrogen fixation.
Line 63-63: Please build on your arguments about why DNRA has a dominant role in nitrate removal.
Line 79: DNA-Stable isotope probing
Line 81: Assimilatory, why is this in quotation marks?
Materials and methods:
2.1 Soil sampling and Characterization
Line 91: Were these samples kept anoxic throughout the samples preparation? If not, this may have had an impact on the microbial communities prior to the start of the experiment.
2.2 Nitrate and nitrous oxide concentration gradient experiments
Line 110: -3.02 µmol/g-dry soil? Surely not a negative value?
Line 120: Citation is gone.
2.3 Soil Microcosms for DNA-SIP Incubation
What % 15N did you use? It is crucial to label your soil with highly enriches 15N (prefrebly >98% 15N)
Line 125: 14N2O/Ar, add isotope in superscript
Line 126: 14N2/Ar, add isotope in superscript
You did not assess atom%?
2.4 Determination of 15N-Labeled Gas 132
I have very limited insights into gas analyses using GC-MS, thus I cannot offer valuable inputs on these analyses, unfortunately.
2.5 DNA Extraction, SIP Gradient Fractionation and Quantitative PCR
Line 150: Please provide the information about the extraction kit and additional relevant information. This includes the buoyant density of the saturated CsCl solution, and the final density of the CsCl+ Gradient buffer (Including DNA).
Line 153: Why is the final buoyant density of the CsCl 1.690 g/ml? The buoyant density of the fractionated samples should be around 1.6 to 1.8 g/ml with an average BD of 1.725. Having a BD at 1.69 will affect your ability to adequately separate labeled and non-labeled DNA. By labeling your system with 15N, which leads to a much lower % increase of MW compared to that of 13C or 18O, makes separation even more challenging. There is no information about the fractionation (I see from fig 2 that it looks like there is a relatively large discrepancy of the number of fractions collected.
Did you use a refractometer when you measured this, and if so which one? Moreover, I would also like to see what your gradient buffer consists of? Surely you added some Tris-HCl, EDTA and maybe even some KCl?
Line 156: You are using a fixed angle rotor and spinning it at 55.000 rpm (I would urge the authors to also add the G force here as well) for 66 hours. The geometry of a fixed angle rotor requires longer spin times compared to that of near vertical/vertical rotors as the labeled DNA needs to travel further to equilibrate. In the literature the numbers I see are usually a bit higher than 66h even for NV/V rotors. What is the reason for not extending the spin time to 72 or even 100H given the challenges associated with angled rotors? The rotor you use can hold 8 tubes, thus you needed to spin several times. Did you randomly assign the different samples to the spins? As you may or may not know there can be batch effect associated with isopynic centrifugation, which may introduce biases.
Buckley et al 2007 ran the experiment with very similar CsCl concentrations, rotor geometry, spin speed and time, and ultimately a similar output of buoyant density/16S copy plots. In this paper the authors collected fractions with densities >1.7 g/ml and spun them again, this resolved the issue with low degree of separation, ultimately allowing the authors to improve their ability compare 14N and 15N labelled microorganisms.
Line 158: There is no information about how you precipitated and cleaned the DNA in the CsCl buffer. Did you use isopropanol or PEG, with or without Glycogen?
Line 159: Please include more detailed description of the qPCR. Why are you targeting such a large fragment, SYBRGreene (and related dyes) perform better with shorter fragments. More information about the standard curves and getting the copy number of 16s would be helpful.
2.6 16S rRNA Gene Amplicon Sequencing
Line 166: The wording of this entire paragraph seems a bit off. No information about primers used, no information about the PCR setup or lib-prep which sequencing platform. This is important information I need to assess the quality of the work. The choice of selecting heavy and light fractions is usually done by a combination of BD assessment and qPCR, not just qPCR alone, which the authors imply. The BD of the collected fractions are much lower than previous studies, which normally is around 1.71-1.72 (ex: see Bell et al. 2011). There is also no information about the removal of rare taxa, rarefaction etc. Where the extraction negatives sequenced, if so, how did the authors deal with any potential contaminants?
There is no information whatsoever about the fractionation. This is a very important step. It is crucial to fractionate the samples as soon as possible after spinning. Did you fractionate one sample at the time, did you use a peristatic pump, how was this done?
The number of fractions collected is different between the samples, this is odd. 11 and 12 for NO3-, 10 and 11 for N2O, and 13 and 12 for N2. I would like to hear the rationale behind this choice. Why did you choose 3 fractions for NO3- and N2 but only 2 in the N2O treatment? The definition of which fractions that there chosen to be sequenced also seems arbitrary, can the authors please explain the rationale behind this?
Moreover, I do not find the supplementary information anywhere, so that’s not good.
2.7 RNA Extraction and Metatranscriptomic analysis
Important information about bioinformatics is largely missing. No information about cDNA synthesis, read depth, what quality control settings were used for QC of reads, bp the clustering etc? This section need a lot of work. There is no version on the silva databases, nor appropriate citations for most of the bioinformatical tools used.
Line 184; More than 10 ug per sample? Some information regarding the concentrations (and variation of these concentrations) would be valuable.
2.8 Statistical Analysis
Why not use DeSeq2 instead of LEFSe? Did you transform your reads in any way?
Results
3.1. Determination of SIP experimental conditions
I cannot access the supplementary data, this it is very difficult for me to address much of the data from this paragraph.
3.2. Inorganic Nitrogen Reduction and 15N-Labeled Gas generation
Line 247: Denitrification should indeed start before 24h why didn’t you measure from the start of the experiment and at 12h as well?
Is there a reason why there is no error bars for all 24H, and the N2O 72h points?
The scales are different on the y-axis
3.3 Quantification of the 16S rRNA gene in CsCl gradient fractions
It’s very confusing that the fractions are named the way they are. H0 to H2 and L1 to L3, moreover why were there only 2 fractions analyzed in the N2O treatment, while in the N2 treatment two fractions are pooled? Why did you Not include the 14N fraction with BD ~1.665 in the N2 Treatment?
This makes interpreting figure 3 much more difficult than it needs to be. Just name the fractions in a meaningful way. Including P-values for the LDA scores is important. Having the same scale on the horizonal axis in fig 3.
Can the authors explain why you just didn’t pool the different fractions together?
There is no mention of GC content of the recovered bacteria in the manuscript whatsoever, this is an immensely important proxy as high GC taxa will naturally merge towards heavier BD, this should be mentioned and discussed in the manuscript.
3.4 Potential 15N-assimilating Microorganisms Revealed by DNA-SIP and 16S rRNA Amplicon Sequencing
I would like there to be some data on the seq id % for some of these OTUs/ASVs. There is no information about which bioinformatic pipelines that were used, which is a concern.
Line 371: Reference is not there
Discussion:
As it stands, this paper fails to present key details about isotopic labeling, library setup, both for the 16S and the metatranscriptomics dataset, bioinformatical and statistical tools. Thus I am not able to accurately assess the quality of the findings, as I am lacking a ton of important information about the data generation. I urge the authors to go through the manuscript and add all key information about the aforementioned points. This is a pity as the questions and outline of the paper is very interesting, using highly relevant techniques, unfortunately there is too much key information missing for me to accurately assess the quality of the work as it stands now.
Comments on the Quality of English LanguageAs a non-native English speaker myself ,I do understand some of the challenges associated with creating a well written and fluid article. Although the paper is decently written I urge the authors to try to improve the fluidity of the language in the manuscript. As it stands now it feels a bit staggered and repetitive.
Author Response
We sincerely thank Reviewer 2 for the constructive reviews. To address the reviewer’s comments, we have revised the manuscript as below. The first draft was proofread by Springer Nature Author Services (A3DF-436B-1882-708D-6263), and the second draft (R1) was proofread by Editage (GNGSD_12). The corresponding certificates for each have been attached.
Introduction
- Line 63: Explain how the presence of pseudo-NifH genes led to misinterpretations in terms of understanding the nitrogen cycle or the interplay between DNRA, denitrification and nitrogen fixation.
→Thank you for your comment. We have clarified how the presence of pseudo-nifH genes can lead to misinterpretations of nitrogen fixation. This explanation has been added to the revised manuscript (Line 63–64).
- Line 63-63: Please build on your arguments about why DNRA has a dominant role in nitrate removal.
→Thank you for your comment. In response to your suggestion, we have expanded the discussion on why DNRA plays a dominant role in nitrate removal in paddy soils. (Line 65–67).
- Line 79: DNA-Stable isotope probing
→Thank you for your comment. The term has been revised accordingly (Line 82).
- Line 81: Assimilatory, why is this in quotation marks?
→Thank you for your comment. Quotation marks were originally used around ‘assimilatory’ to distinguish it from dissimilatory processes; however, they have been removed in the revised manuscript in response to this comment (Line 84).
Materials and methods:
2.1 Soil sampling and Characterization
- Line 91: Were these samples kept anoxic throughout the samples preparation? If not, this may have had an impact on the microbial communities prior to the start of the experiment.
→Thank you for your comment. Yes, an anoxic environment was maintained throughout the entire process. After removing impurities and homogenizing the samples, anaerobic pre-incubation was initiated immediately, and all subsequent steps were carried out under anoxic conditions.
2.2 Nitrate and nitrous oxide concentration gradient experiments
- Line 110: -3.02 µmol/g-dry soil? Surely not a negative value?
→Thank you for pointing this out. This was a typographical error. We apologize for the mistake and have corrected it in the revised manuscript (Line 112).
- Line 120: Citation is gone.
→Thank you for your comment. We have added the missing citation in the revised version (Line 121-122).
2.3 Soil Microcosms for DNA-SIP Incubation
- What % 15N did you use? It is crucial to label your soil with highly enriches 15N (prefrebly >98% 15N)
→Thank you for your comment. We used pure 15N2, 15N2O (> 99.9%; GL Sciences, Inc. Tokyo, Japan), and Na15NO3 (> 99.8%; SI Science, Saitama, Japan). We have added the missing information in the revised manuscript (Line 133-134).
- Line 125: 14N2O/Ar, add isotope in superscript
- Line 126: 14N2/Ar, add isotope in superscript
→Thank you for your comment. These modifications have been made (Line 131-132).
- You did not assess atom%?
→Thank you for your comment. Figure 1 has been revised accordingly.
2.4 Determination of 15N-Labeled Gas 132
- I have very limited insights into gas analyses using GC-MS, thus I cannot offer valuable inputs on these analyses, unfortunately.
→We sincerely appreciate your time and effort in reviewing our manuscript.
2.5 DNA Extraction, SIP Gradient Fractionation and Quantitative PCR
- Line 150: Please provide the information about the extraction kit and additional relevant information. This includes the buoyant density of the saturated CsCl solution, and the final density of the CsCl+ Gradient buffer (Including DNA).
→Thank you for your comment. We have added information about the DNA extraction and the buoyant density of the saturated CsCl solution. The volume of the added DNA solution was very small and therefore had a negligible effect on the final density of the CsCl gradient buffer (Line 158–168).
- Line 153: Why is the final buoyant density of the CsCl 1.690 g/ml? The buoyant density of the fractionated samples should be around 1.6 to 1.8 g/ml with an average BD of 1.725. Having a BD at 1.69 will affect your ability to adequately separate labeled and non-labeled DNA. By labeling your system with 15N, which leads to a much lower % increase of MW compared to that of 13C or 18O, makes separation even more challenging. There is no information about the fractionation (I see from fig 2 that it looks like there is a relatively large discrepancy of the number of fractions collected.
→Thank you for your comment. Prior to the main experiment, we conducted preliminary tests using CsCl gradients with different final buoyant densities. Based on these results, a final density of 1.690 g/ml provided the most effective separation for our specific samples and labeling conditions. To further enhance resolution and ensure adequate separation of labeled and unlabeled DNA—particularly given the relatively small mass difference introduced by 15N labeling—we increased the number of fractions collected after ultracentrifugation. This information has been added to the revised manuscript (Line 171–174). Regarding the variation in the number of fractions showing labeled signals across treatments (as shown in Fig. 2), we believe this is due to differences in microbial community composition among the different amendment groups.
- Did you use a refractometer when you measured this, and if so which one? Moreover, I would also like to see what your gradient buffer consists of? Surely you added some Tris-HCl, EDTA and maybe even some KCl?
→Thank you for your comment. Yes, we used a refractometer (PR-RI ,ATAGO, Co. Ltd., Tokyo, Japan)., and the relevant information has been added to the revised manuscript (Line 175).
As for the gradient buffer, it was prepared by combining 50 ml of 1 M Tris-HCl, 3.75 g of KCl, and 1 ml of 0.5 M EDTA in 400 ml of distilled water. After dissolving the KCl, the volume was adjusted to 500 ml with ddH₂O. The solution was then filter-sterilized and autoclaved. The final composition of the gradient buffer was 0.1 M Tris, 0.1 M KCl, and 1 mM EDTA.
- Line 156: You are using a fixed angle rotor and spinning it at 55.000 rpm (I would urge the authors to also add the G force here as well) for 66 hours. The geometry of a fixed angle rotor requires longer spin times compared to that of near vertical/vertical rotors as the labeled DNA needs to travel further to equilibrate. In the literature the numbers I see are usually a bit higher than 66h even for NV/V rotors. What is the reason for not extending the spin time to 72 or even 100H given the challenges associated with angled rotors? The rotor you use can hold 8 tubes, thus you needed to spin several times. Did you randomly assign the different samples to the spins? As you may or may not know there can be batch effect associated with isopynic centrifugation, which may introduce biases.
→Thank you for your insightful comment regarding the ultracentrifugation conditions. We agree that the geometry of fixed-angle rotors requires careful consideration when optimizing separation efficiency. In our case, the 66-hour centrifugation at 55,000 rpm was selected based on prior literature (Buckley D.H., Huangyutitham V., Hsu S.F., et al. Stable isotope probing with 15N₂ reveals novel noncultivated diazotrophs in soil. Appl Environ Microbiol, 2007, 73(10): 3196–3204) and our preliminary trials, which demonstrated that labeled DNA formed clearly resolved bands under these conditions.
Nonetheless, we acknowledge the importance of rotor geometry and have now included the calculated relative centrifugal force (RCF, 172,750×g) to clarify the effective separation power applied (Line 169). We believe that the chosen duration represents a practical balance between resolution, sample integrity, and centrifuge safety constraints.
We are aware of the potential batch effects associated with isopycnic centrifugation. Therefore, instead of randomly assigning samples to spins, we carefully processed experimental and control samples from the same treatment group within the same centrifugation run. To further minimize potential biases, the buoyant density and refractive index of each fraction were measured and compared against standard solutions after each run to ensure consistency across samples.
- Buckley et al 2007 ran the experiment with very similar CsCl concentrations, rotor geometry, spin speed and time, and ultimately a similar output of buoyant density/16S copy plots. In this paper the authors collected fractions with densities >1.7 g/ml and spun them again, this resolved the issue with low degree of separation, ultimately allowing the authors to improve their ability compare 14N and 15N labelled microorganisms.
→Thank you for your comment. In our study, we chose a different strategy based on our preliminary tests. Specifically, we optimized the initial buoyant density and increased the number of fractions collected (25 per tube) to enhance resolution.
- Line 158: There is no information about how you precipitated and cleaned the DNA in the CsCl buffer. Did you use isopropanol or PEG, with or without Glycogen?
→Thank you for your comment. We have now added detailed information regarding DNA precipitation and purification. DNA recovered from each fraction was precipitated using a polyethylene glycol solution (30% PEG, 1.6 M NaCl) with 20 µg of glycogen, washed with 70% ethanol, and then resuspended in 30 µL of TE buffer (pH 8.0). These details have been included in the revised Methods section (Line 176–178).
- Line 159: Please include more detailed description of the qPCR. Why are you targeting such a large fragment, SYBRGreene (and related dyes) perform better with shorter fragments. More information about the standard curves and getting the copy number of 16s would be helpful.
→Thank you for your comment. We have added more detailed information on the qPCR procedure. The 27F/520R primer set has been previously validated in our related studies for quantifying bacterial abundance in density gradient fractions (Zhang et al., 2023, Active Nitrogen Fixation by Iron-Reducing Bacteria in Rice Paddy Soil and Its Further Enhancement by Iron Application). While it is true that SYBR Green-based assays generally perform optimally with shorter amplicons, this primer set demonstrated sufficient amplification efficiency and specificity in our system. Standard curves were generated using serial dilutions of standard DNA, and 16S rRNA gene copy numbers were calculated based on Ct values and the slope of the standard curve (Line 182–184).
2.6 16S rRNA Gene Amplicon Sequencing
- Line 166: The wording of this entire paragraph seems a bit off. No information about primers used, no information about the PCR setup or lib-prep which sequencing platform. This is important information I need to assess the quality of the work. The choice of selecting heavy and light fractions is usually done by a combination of BD assessment and qPCR, not just qPCR alone, which the authors imply. The BD of the collected fractions are much lower than previous studies, which normally is around 1.71-1.72 (ex: see Bell et al. 2011). There is also no information about the removal of rare taxa, rarefaction etc. Where the extraction negatives sequenced, if so, how did the authors deal with any potential contaminants?
We have revised the corresponding paragraph in the manuscript to include detailed information on the primers used, PCR conditions, library preparation, and sequencing platform. These additions aim to improve clarity and allow for a more thorough evaluation of data quality.
→Thank you for your comment. We acknowledge that the buoyant densities (BDs) of the collected fractions are somewhat lower than the typical range (e.g., 1.71–1.72) reported in previous DNA-SIP studies (e.g., Bell et al., 2011). Several factors may contribute to this discrepancy. Sample-specific characteristics, such as the GC content and size distribution of DNA within our soil microbial communities, can affect the equilibrium density during ultracentrifugation. Additionally, the incubation period and the degree of 15N incorporation in our experiment may have been lower than those in other studies. Nevertheless, we observed a clear shift in DNA abundance and microbial community composition toward the heavier fractions in the15N-labeled treatments compared to the controls, indicating successful isotopic labeling and separation. We therefore believe that the observed BD range is consistent with our experimental design and microbial system and still allows for valid interpretation of isotopic enrichment.
We appreciate the reviewer’s comments regarding data preprocessing and contamination control. In our study, we did not perform rarefaction or explicitly remove rare taxa. Instead, we relied on the default quality filtering and denoising workflow implemented in the DADA2 plugin within QIIME2, which is designed to remove low-quality, low-frequency, and potentially spurious sequences.
Regarding contamination control, although extraction negative controls were not included in the sequencing run, we took extensive precautions during DNA extraction and PCR setup to minimize the risk of contamination, including the use of sterilized equipment, aerosol-resistant filter tips, and dedicated clean workspace areas. Moreover, no amplification was observed in the negative PCR controls, indicating minimal risk of reagent-derived contamination.
We acknowledge the importance of these considerations, and we believe that the quality control measures, and handling procedures applied in this study support the reliability of the sequencing data obtained.
- There is no information whatsoever about the fractionation. This is a very important step. It is crucial to fractionate the samples as soon as possible after spinning. Did you fractionate one sample at the time, did you use a peristatic pump, how was this done?
→Thank you for your comment. We apologize for the omission of detailed information regarding the fractionation process and have now added this information. The protocol we followed was based on a previous study (Dunford E.A., Neufeld J.D. DNA stable-isotope probing (DNA-SIP), Journal of Visualized Experiments: JoVE, 2010 (42): 2027). After ultracentrifugation, each gradient tube was immediately fractionated individually using a syringe pump from the bottom of the tube. To minimize gradient diffusion and preserve resolution, all samples were processed within two hours after centrifugation (Line 174-178).
- The number of fractions collected is different between the samples, this is odd. 11 and 12 for NO3-, 10 and 11 for N2O, and 13 and 12 for N2. I would like to hear the rationale behind this choice. Why did you choose 3 fractions for NO3- and N2 but only 2 in the N2O treatment? The definition of which fractions that there chosen to be sequenced also seems arbitrary, can the authors please explain the rationale behind this?
→Thank you for your comment. The number of fractions shown in the figure differs among treatments because only the major fractions within a buoyant density range of 1.63–1.71 g/ml were selected to enhance visual clarity and facilitate comparison among treatments. All samples were fractionated into 25 fractions, and the number of fractions displayed does not reflect the total number collected. The displayed fractions were chosen based on where the most informative shifts in DNA abundance or labeling occurred, and this did not affect the overall interpretation of the results (Line 171-174).
The number of sequenced fractions differed among treatments because fractions were selected based on whether there was a meaningful contrast between the treatment and control samples at the same buoyant density. In the N₂O treatment, only two fractions showed clear differences between labeled and unlabeled samples at the same density, whereas in the NO₃⁻ and N₂ treatments, three such fractions were identified.
- Moreover, I do not find the supplementary information anywhere, so that’s not good.
→Thank you for your comment. The supplementary information was submitted along with the manuscript; however, it may not have been properly linked or accessible during the review process. We will contact the editorial office to ensure that the supplementary files are properly shared with the reviewers.
2.7 RNA Extraction and Metatranscriptomic analysis
- Important information about bioinformatics is largely missing. No information about cDNA synthesis, read depth, what quality control settings were used for QC of reads, bp the clustering etc? This section need a lot of work. There is no version on the silva databases, nor appropriate citations for most of the bioinformatical tools used.
→Thank you for your comment. We have added Supplemental information (Table S5).
- Line 184; More than 10 ug per sample? Some information regarding the concentrations (and variation of these concentrations) would be valuable.
→Thank you for your comment. We have added Supplemental information (Table S4).
2.8 Statistical Analysis
- Why not use DeSeq2 instead of LEFSe? Did you transform your reads in any way?
→Thank you for your suggestion regarding the use of alternative biomarker detection tools such as DESeq2. We agree that these methods are powerful and statistically robust for differential abundance analysis. However, we chose to use LEfSe in this study because it remains one of the most widely used and well-established tools in microbiome research, particularly for biomarker discovery. As discussed in Nearing et al. (2022) [https://doi.org/10.1186/s12859-021-04193-6], the results obtained using LEfSe and ANCOM-BC are often highly concordant. This suggests that, despite some methodological differences, LEfSe can still provide biologically meaningful and reliable results that align well with more recent analytical approaches. Moreover, LEfSe offers the advantages of intuitive interpretability and broad acceptance in the field, which facilitates comparison with previously published studies. Considering these factors, along with the overall consistency of our findings with known microbial responses, we respectfully prefer to retain LEfSe for the differential abundance analysis in this study and do not intend to revise the corresponding figures.
Results
3.1. Determination of SIP experimental conditions
- I cannot access the supplementary data, this it is very difficult for me to address much of the data from this paragraph.
→Thank you for your comment. The supplementary information was submitted along with the manuscript; however, it may not have been properly linked or accessible during the review process. We will contact the editorial office to ensure that the supplementary files are properly shared with the reviewers.
3.2. Inorganic Nitrogen Reduction and 15N-Labeled Gas generation
- Line 247: Denitrification should indeed start before 24h why didn’t you measure from the start of the experiment and at 12h as well?
→Thank you for your comment. We acknowledge that denitrification may begin earlier than 24 hours. However, our primary objective was to assess isotopic incorporation into microbial DNA, which generally requires sufficient time for assimilation. Preliminary trials also indicated that isotope labeling signals were too weak to be reliably detected before 24 hours. Therefore, we focused our sampling on later time points to ensure meaningful interpretation of both functional activity and community-level changes.
- Is there a reason why there is no error bars for all 24H, and the N2O 72h points?
The scales are different on the y-axis
→Thank you for your comment. Error bars are included; however, they are not visually prominent due to the minimal variation among replicates. To better reflect differences in concentrations and improve the clarity of data visualization and interpretation, the y-axis scales were adjusted individually for each treatment group.
3.3 Quantification of the 16S rRNA gene in CsCl gradient fractions
- It’s very confusing that the fractions are named the way they are. H0 to H2 and L1 to L3, moreover why were there only 2 fractions analyzed in the N2O treatment, while in the N2 treatment two fractions are pooled? Why did you Not include the 14N fraction with BD ~1.665 in the N2 Treatment?
→Thank you for your comment. We acknowledge that the naming of fractions (H0–H2 and L1–L3) may be confusing; however, we were unable to find more suitable alternatives. In our naming convention, “H” and “L” refer to fractions from the heavy and light regions of the density gradient, respectively, with the numbers used for simple identification. For the N₂O treatment, only two fractions were analyzed because these were the only ones that showed meaningful differences between the treatment and control samples at the same DNA buoyant density. In the N₂ treatment, two fractions were pooled because their buoyant densities were very close, and it was more appropriate to combine them for comparison with the corresponding 14N fractions. Moreover, previous DNA-SIP studies using 15N₂, such as Zhang et al. (2023), have shown that key nitrogen-fixing microorganisms tend to be enriched not only around the peak buoyant density but also in heavier fractions. Based on this evidence, for the N₂ treatment, we selected the peak fraction along with the adjacent heavier and lighter fractions, excluding the much lighter fraction with a buoyant density around 1.665 g/ml.
- This makes interpreting figure 3 much more difficult than it needs to be. Just name the fractions in a meaningful way. Including P-values for the LDA scores is important. Having the same scale on the horizonal axis in fig 3.
→Thank you for your comment. Regarding the naming of fractions in Figure 3, we carefully considered common practices but did not identify a more meaningful or widely accepted alternative to the current labels. Therefore, we retained them to maintain clarity and consistency. To indicate statistical significance, p-values for the LDA scores have been added. Additionally, the horizontal axes in all panels of Figure 3 have been standardized to the same scale to facilitate direct comparison.
- Can the authors explain why you just didn’t pool the different fractions together?
→Thank you for your comment. We did not pool the fractions because each density fraction represents a distinct position within the DNA gradient, corresponding to different levels of 15N incorporation. Pooling the fractions could obscure differences in microbial community composition and subtle variations in labeling intensity, thereby reducing the resolution needed to identify specific active microbes. Analyzing the fractions separately allows for a more precise understanding of nitrogen isotope incorporation dynamics and microbial functional diversity. This approach ensures accurate interpretation of the data and preserves the biological significance of the results.
- There is no mention of GC content of the recovered bacteria in the manuscript whatsoever, this is an immensely important proxy as high GC taxa will naturally merge towards heavier BD, this should be mentioned and discussed in the manuscript.
→Thank you for your comment. We agree that GC content can influence the buoyant density of DNA and should therefore be considered when interpreting DNA-SIP results. However, in our study, microbial communities were compared between 14N and 15N-labeled samples within the same buoyant density fractions. This design minimizes the confounding effects of GC content, as any GC-dependent shifts would similarly affect both 14N and 15N samples at a given density. Moreover, we did not rely on a single density fraction to identify labeled taxa. Instead, we analyzed multiple fractions across the gradient and focused on consistent differences between treatment and control groups. This approach enhances confidence that the observed community shifts reflect isotopic labeling rather than intrinsic factors such as GC content.
3.4 Potential 15N-assimilating Microorganisms Revealed by DNA-SIP and 16S rRNA Amplicon Sequencing
- I would like there to be some data on the seq id % for some of these OTUs/ASVs. There is no information about which bioinformatic pipelines that were used, which is a concern.
→Thank you for your comment. Due to the large number of ASVs generated by high-throughput sequencing, it is impractical to provide sequence identity percentages for each individual OTU/ASV. Instead, taxonomic assignments were made using established 16S rRNA gene sequence identity thresholds as described by Yarza et al. (2014), and the overall classification results are summarized in Supplementary Table S3. In addition, detailed information about the bioinformatic pipeline has been added to the Methods section (Line 199-209).
Line 371: Reference is not there
→Thank you for your comment. The reference has been added (Line 407-409).
Discussion:
As it stands, this paper fails to present key details about isotopic labeling, library setup, both for the 16S and the metatranscriptomics dataset, bioinformatical and statistical tools. Thus I am not able to accurately assess the quality of the findings, as I am lacking a ton of important information about the data generation. I urge the authors to go through the manuscript and add all key information about the aforementioned points. This is a pity as the questions and outline of the paper is very interesting, using highly relevant techniques, unfortunately there is too much key information missing for me to accurately assess the quality of the work as it stands now.
→We sincerely thank you for your constructive comments and for highlighting the scientific relevance of our study. We acknowledge that the initial manuscript lacked sufficient methodological detail, which may have hindered a thorough assessment of the data quality and reproducibility. In the revised version, we have thoroughly updated the Materials and Methods section to provide clear and detailed descriptions. In addition, we have included relevant parameters, software versions, and references where appropriate. We believe these revisions significantly improve the clarity and rigor of the manuscript, and we hope that the updated version addresses your concerns. We are grateful for the opportunity to improve our manuscript.
Reviewer 3 Report
Comments and Suggestions for Authors
The study entitled “Ammonium Generating Microbial Consortia via Reductive Nitrogen Transformation in Paddy Soil Revealed by the Combination of DNA-Stable Isotope Probing and Metatranscriptomics” presents a timely investigation combining isotope probing and metatranscriptomics. It is indeed very deep study encompassing DNA-SIP and metatranscriptomics. However, there are major issues that the authors need to address.
Major
- The authors utilized LEfSe for differential abundance analysis, however, I recommend using the most powerful biomarker tools: ANCOM-BC and ALDEx2, which are considered more reliable and powerful.
- Please clearly indicate the type of experimental design used in the study.
Minor
-Typos and grammatical errors
-Avoid slashes and consider consistent use of hyphens
Author Response
We sincerely thank reviewer 3 for the positive and detailed reviews. We have revised the manuscript to improve the readability as per the reviewer’s suggestions. The first draft was proofread by Springer Nature Author Services (A3DF-436B-1882-708D-6263), and the second draft (R1) was proofread by Editage (GNGSD_12). The corresponding certificates for each have been attached.
Major
- The authors utilized LEfSe for differential abundance analysis, however, I recommend using the most powerful biomarker tools: ANCOM-BC and ALDEx2, which are considered more reliable and powerful.
→Thank you for your suggestion regarding the use of alternative biomarker detection tools such as ANCOM-BC and ALDEx2. We agree that these methods are powerful and statistically robust for differential abundance analysis.
However, we chose to use LEfSe in this study because it remains one of the most widely used and well-established tools in microbiome research, particularly for biomarker discovery. As noted in Nearing et al. (2022) [https://doi.org/10.1186/s12859-021-04193-6], the results obtained using LEfSe and ANCOM-BC are often highly concordant, suggesting that despite methodological differences, LEfSe can still yield biologically meaningful and reliable results that align well with more recent approaches. Additionally, LEfSe offers the advantages of being easily interpretable and widely accepted in the field, facilitating comparison with previously published studies. Given these considerations and the overall consistency of our results with known microbial responses, we respectfully prefer to retain LEfSe for differential abundance analysis in this study and do not intend to revise the figures accordingly.
- Please clearly indicate the type of experimental design used in the study.
→Thank you for your valuable suggestion. In the revised manuscript, we have added a clear description of the experimental design in Section 2.3. Specifically, we now state that a randomized controlled experimental design was employed, consisting of three 15N-labeled nitrogen treatment groups (+15NO₃⁻, +15N₂O, +15N₂) and one soil-only control group (CK). Each group included three replicates to ensure the reliability of the results and the validity of statistical analyses.
Minor
- -Typos and grammatical errors
-Avoid slashes and consider consistent use of hyphens
→Thank you for your careful review and valuable suggestions. We have carefully reviewed the manuscript, correcting all typographical and grammatical errors. In addition, we have revised the manuscript to avoid the use of slashes and have ensured consistent use of hyphens or rephrased the relevant expressions for clarity and consistency.
Reviewer 4 Report
Comments and Suggestions for Authors
Comment:
This study employed radioactive isotope labeling to trace nitrogen metabolism. Initially, the optimal calibration time was determined, followed by analysis of 15N-assimilated 16S rDNA to identify nitrogen-assimilating bacteria. Subsequently, transcriptomic analysis was conducted to investigate key enzymes involved in nitrogen metabolism and their associated bacterial taxa. The research methods and results are presented clearly and accurately.
Suggestions:
Including nitrogen fertilizer treatments and time-course variables may help to reveal the dynamic changes in soil nitrogen-metabolizing bacterial communities.
In Figure 3, the labeled bacteria contain radioactive nitrogen in their DNA, indicating active intracellular nitrogen metabolism. These bacteria are identified at the genus level. In contrast, Figure 4 presents transcriptomic data linking nitrogen metabolism-related genes to bacteria, labeled at the family level. Clarification on the relationship between these two datasets is recommended. Specifically, it would be helpful to indicate whether the genera identified in Figure 3 can be taxonomically matched to the families shown in Figure 4.
Author Response
We sincerely thank reviewer 4 for the constructive reviews. We have revised the manuscript according to the reviewer’s suggestions.
Suggestions:
Including nitrogen fertilizer treatments and time-course variables may help to reveal the dynamic changes in soil nitrogen-metabolizing bacterial communities.
→We appreciate the reviewer’s suggestion. However, the primary objective of this study was to identify the microbial consortia responsible for ammonium generation in paddy soils—a process that has not yet been clearly characterized. While we agree that fertilizer treatments could provide important insights into microbial community responses under agronomic conditions, such investigations are beyond the scope of this study and represent a valuable direction for future research.
Regarding the time-course design, we fully agree that temporal dynamics are crucial for understanding microbial processes. However, our study specifically aimed to identify microbes actively involved in ammonium generation. Extending the incubation period might result in more complex nitrogen transformation patterns due to potential cross-feeding and secondary utilization of labeled nitrogen, which may obscure the direct associations between specific microbes and ammonium production. To avoid such confounding effects, we intentionally employed a short-term incubation strategy to more precisely capture the initial nitrogen transformation events with higher specificity and resolution.
In Figure 3, the labeled bacteria contain radioactive nitrogen in their DNA, indicating active intracellular nitrogen metabolism. These bacteria are identified at the genus level. In contrast, Figure 4 presents transcriptomic data linking nitrogen metabolism-related genes to bacteria, labeled at the family level. Clarification on the relationship between these two datasets is recommended. Specifically, it would be helpful to indicate whether the genera identified in Figure 3 can be taxonomically matched to the families shown in Figure 4.
→We appreciate the reviewer’s comment. The difference in taxonomic resolution between Figure 3 and Figure 4 reflects the inherent characteristics of the data obtained from each approach. In Figure 3, the 16S rRNA gene-based DNA-SIP analysis allowed taxonomic identification at the genus level. In contrast, the transcriptomic data presented in Figure 4 were limited to the family level in terms of resolution. To clarify the relationship between these datasets, we have included an explanation in our revised manuscript. Specifically, we noted that many genera identified by DNA-SIP (e.g., Geobacter, Bacillus, Clostridium) are phylogenetically affiliated with the families detected in the transcriptomic analysis (e.g., Geobacteraceae, Bacillaceae, Clostridiaceae).
Reviewer 5 Report
Comments and Suggestions for Authors
This manuscript investigates the microbial consortia responsible for ammonium generation in paddy soils via three key reductive nitrogen transformation processes: dissimilatory nitrate reduction to ammonium (DNRA), denitrification, and nitrogen fixation. The study uses a well-designed combination of 15N-DNA stable isotope probing (SIP) and metatranscriptomic analyses in controlled microcosm experiments with labeled substrates (15NO₃⁻, 15N₂O, 15N₂). It identifies key bacterial families contributing to extracellular and intracellular ammonium generation under different nitrogen source conditions. Notably, Geobacteraceae, Bacillaceae, Clostridiaceae, Anaeromyxobacteraceae, and Rhodocyclaceae are consistently highlighted as active contributors across multiple nitrogen pathways. The study emphasizes the ecological relevance of these findings in the context of sustainable nitrogen management in rice paddies.
Comments
Strengths:
The research question is timely, relevant, and well-formulated.
The integration of 15N-SIP and metatranscriptomics is innovative and enables a powerful functional–taxonomic mapping of nitrogen transformation processes.
Methodology is comprehensive and rigorously executed, including appropriate controls and time-point sapling.
The manuscript is well-organized, with clear narrative flow from introduction to discussion.
Results are preented with appropriate figures and tables, and the findings are logically interpreted.
Weaknesses:
There are minor but consistent issues in English grammar and phrasing throughout the manuscript.
Some statements in the discussion section are speculative without direct supporting data (e.g., line 415).
Figure legends and supplementary material references occasionally shows mistakes (e.g., “[Error! Reference source not found.]” in lines 119, 371).
Statistical analyses of metatranscriptomic data could be more deeply discussed to validate observed differences.
Line 2–5: The title is informative but quite long; consider shortening or restructuring for clarity.
Lines 62–67: The challenge of studying pseudo-nifH genes is briefly noted—consider expanding slightly to underscore its implications for past research misinterpretations.
Line 119: “Error! Reference source not found.” must be corrected.
Lines 164–174: Clarify primer sequences and region for 16S rRNA sequencing; include more details or move Table S2 reference to main text. Please, give more details about sampling, number and replicate for each analysis you performed (DNA sequencing, metatrascriptomic, etc.)
Lines 177–186: Include library preparation kits and protocols used for metatranscriptomics.
Lines 247–256: In Figure 1, define abbreviations in the legend (e.g., v/v %).
Line 264–275: Strong qPCR support for 15N incorporation; suggest plotting absolute 16S gene copy numbers as a complement to relative abundance.
Lines 371 and 415: Two “Error! Reference source not found.”
Lines 387–394: The time-lag explanation for discrepancies between transcriptomics and SIP is reasonable but would benefit from a citation or prior example.
Line 414: The statement on alternative N₂O-to-nitrate/nitrite pathways is intriguing but speculative. Recommend softening language or providing direct evidence or a referenced mechanism.
Line 470: The text refers to "Figure 3", but no such figure is provided—ensure numbering and placement of all figures are correct.
Author Response
We sincerely thank the reviewer 5 for their positive and encouraging feedback. We edited the content based on the comments below. The first draft was proofread by Springer Nature Author Services (A3DF-436B-1882-708D-6263), and the second draft (R1) was proofread by Editage (GNGSD_12). The corresponding certificates for each have been attached.
Weaknesses:
There are minor but consistent issues in English grammar and phrasing throughout the manuscript.
- Some statements in the discussion section are speculative without direct supporting data (e.g., line 415).
→Thanks for your comment. As suggested, we have revised the sentence (Line 476-478).
- Figure legends and supplementary material references occasionally shows mistakes (e.g., “[Error! Reference source not found.]” in lines 119, 371).
→Thanks for your comment. We have corrected these reference sources (Lines 153 and 432).
- Statistical analyses of metatranscriptomic data could be more deeply discussed to validate observed differences.
→Thanks for your suggestion. To better represent differences in gene expression levels, we revised the figure using statistically more robust TPM values instead of relative abundance ratios based on 16S rRNA (Figure 4b, c).
- Line 2–5: The title is informative but quite long; consider shortening or restructuring for clarity.
→Thanks for your suggestion. We’ve changed the title as “Ammonium Generating Microbial Consortia in Paddy Soil Revealed by DNA-Stable Isotope Probing and Metatranscriptomics”.
- Lines 62–67: The challenge of studying pseudo-nifH genes is briefly noted—consider expanding slightly to underscore its implications for past research misinterpretations.
→Thanks for your comment. We have revised it to further elaborate on the challenges posed by pseudo-nifH genes and their potential to cause misinterpretation in past studies (Line 63-64).
- Line 119: “Error! Reference source not found.” must be corrected.
→We have corrected the reference sources (Line 153).
- Lines 164–174: Clarify primer sequences and region for 16S rRNA sequencing; include more details or move Table S2 reference to main text. Please, give more details about sampling, number and replicate for each analysis you performed (DNA sequencing, metatrascriptomic, etc.)
→Thanks for your comment. We have clarified the primer sequences and target region for 16S rRNA sequencing in Method 2.6 and added more details regarding the sampling strategy, number of replicates, and experimental design for each analysis (including DNA-SIP and metatranscriptomics).
- Lines 177–186: Include library preparation kits and protocols used for metatranscriptomics.
→Thanks for your comment. We’ve added the sentence below in Lines 225-229.
- Lines 247–256: In Figure 1, define abbreviations in the legend (e.g., v/v %).
→Thanks for your comment. The abbreviation “v/v %” has been defined in the revised figure legend of Figure 1.
- Line 264–275: Strong qPCR support for 15N incorporation; suggest plotting absolute 16S gene copy numbers as a complement to relative abundance.
→Thanks for the valuable suggestion. We have added a plot of absolute 16S rRNA gene copy numbers to complement the relative abundance data. This information is now included in Figure S5 and described in Results section 3.3 of the revised manuscript (Line 322-323).
- Lines 371 and 415: Two “Error! Reference source not found.”
→Thanks for your comment. We have corrected the reference sources.
- Lines 387–394: The time-lag explanation for discrepancies between transcriptomics and SIP is reasonable but would benefit from a citation or prior example.
→Thanks for your comment. We have added appropriate citations to support the explanation regarding the time lag between RNA expression and DNA labeling. These references have been included in the revised manuscript (Line 451).
- Line 414: The statement on alternative N₂O-to-nitrate/nitrite pathways is intriguing but speculative. Recommend softening language or providing direct evidence or a referenced mechanism.
→Thanks for your suggestion. We’ve changed the description (Lines 476-478).
- Line 470: The text refers to "Figure 3", but no such figure is provided—ensure numbering and placement of all figures are correct.
→Thanks for your careful observation. We have corrected the figure numbering (Lines 485).